# Identifying a predictive level of serum C-terminal telopeptide associated with a low risk of medication-related osteonecrosis of the jaw secondary to oral surgery: A systematic review and meta-analysis

Camille Ghio[ID][1], Robinson Gravier-Dumonceau[2], Pierre Lafforgue[1], Roch Giorgi[2], Thao Pham[ID][1]*

**1** Department of Rheumatology, Aix Marseille Univ, APHM, Hôpital Sainte-Marguerite, Marseille, France, **2** Aix Marseille Univ, APHM, INSERM, IRD, SESSTIM, Sciences Economiques & Sociales de la Santé & Traitement de l'Information Médicale, ISSPAM, Hop Timone, BioSTIC, Biostatistique et Technologies de l'Information et de la Communication, Marseille, France

* thao.pham@ap-hm.fr

## Abstract

### Objective

Our aim was to determine serum C-terminal telopeptide of type I collagen (sCTX) thresholds for predicting the minimal risk of medication-related osteonecrosis of the jaw (MRONJ) in patients undergoing anti-resorptive therapy prior to oral surgery.

### Methods

A systematic literature search was conducted in MEDLINE, EMBase, and the Cochrane Library up to September 2023 for case-control, prospective and retrospective studies that assessed sCTX levels in patients exposed to anti-resorptive drugs who underwent oral surgery. We extracted data using a predetermined form. We performed an original percentile meta-analysis method, following PRISMA-DTA guidelines and descriptive analysis to identify the threshold associated with the lowest risk while assessing the overall result of the 95th, 97.5th and 99th percentiles with a random-effect model with weighting by DerSimonian and Laird (RStudio software [v. 4.2.0]).

### Results

Seven studies involving 1281 patients were included. Most patients (96%) were treated for osteoporosis, predominantly with oral bisphosphonates (94.5%). Individual data were available for 58 patients. In the entire population of patients who experienced MRONJ after oral surgery (n = 113), the 95th, 97.5th and 99th percentiles of

**Data availability statement:** All relevant data are within the paper and its Supporting Information files.

**Funding:** The author(s) received no specific funding for this work.

**Competing interests:** The authors have declared that no competing interests exist.

sCTX were 338.0 pg/mL [95%CI: 190,3; 485,7], 401.9 pg/mL [95%CI: 191,3; 612,6], and 458.0 pg/mL [95%CI: 190,4; 725,6], respectively.

Among those treated with oral bisphosphonates for osteoporosis (n = 38), the sCTX 95th, 97.5th and 99th percentiles were 185.3 pg/mL [95%CI: 131,3; 239,3] 187.4 pg/mL [95%CI: 133,9; 240,8] and 188.6 pg/mL [95%CI: 135,4; 241,9], respectively. The determination of these same percentiles with individual data analysis yielded similar results, i.e., 202.0, 257.0 and 260.0 pg/mL.

## Conclusion

This pioneering meta-analysis assesses the risk of MRONJ by analyzing sCTX levels in patients undergoing oral surgery while exposed to antiresorptive drugs. Among patients receiving oral bisphosphonate therapy for osteoporosis, a sCTX threshold of 260 pg/mL is linked to an extremely low risk of MRONJ occurrence, surpassing the 99th percentile. Conversely, for patients undergoing treatment for cancer-related conditions, sCTX levels do not reliably serve as a biomarker for identifying this risk.

## Introduction

Antiresorptive drugs (ARD) including bisphosphonates and denosumab, have demonstrated efficacy in osteoporosis management, both in primary and secondary fracture prevention, as well as in reducing associated mortality [1–4]. They have also demonstrated efficacy in cancer-related conditions, i.e., metastatic bone lesions and multiple myeloma lesions.

However, the use of ARD for osteoporosis has seen a decline since the 2010s [5–8]. This decrease can, among other things, be attributed to concerns regarding adverse events associated with anti-resorptive therapies, notably osteonecrosis of the jaw [9,10], even though this side effect is rare and does not challenge the overall positive risk-benefit balance of these treatments [11–14].

Medication-related osteonecrosis of the jaws (MRONJ) cases have been first reported with bisphosphonates in 2003 and similar cases have been documented with denosumab. Since then, osteonecrosis of the jaw has also been reported in patients in the absence of antiresorptive therapy intake. [15–17]. Factors associated with MRONJ include underlying bone disease, dosage and frequency of ARD therapy, smoking, poor oral hygiene, local infections and dental procedures involving bone intrusion [14,18,19]. The incidence of MRONJ is 10 times higher in cancer patients compared to osteoporotic patients. Although rare, this condition is considered a serious adverse event, leading to several clinical issues, including pain and discomfort, difficulty in eating and speaking and infection. The chronic pain, functional impairments, and visible bone exposure associated with this condition can significantly impact the patient's quality of life.

Therefore, there is interest in identifying biomarkers capable of assessing the risk of MRONJ in a patient undergoing ARD therapy before receiving dental care. Among potential biomarkers, serum C-terminal telopeptide cross-link of type 1

collagen (sCTX) is the most studied bone turnover marker [20–23]. From a population of 30 consecutive patients exposed to bisphosphonates who developed MRONJ after oral surgery, Marx et al. were the first to propose risk thresholds: the risk was high if sCTX levels were below 100 pg/mL, moderate if levels were between 100 and 150 pg/mL, and low above 150 pg/mL. However, subsequent studies have not confirmed the protective character of the 150 mg/mL threshold for sCTX, without other thresholds being assessed [24].

Our aim was to determine a safety sCTX threshold capable of assessing the lowest risk of MRONJ in a patient undergoing anti-resorptive therapy prior to dental care through a systematic review and meta-analysis.

## Methods

### Protocol and registration

The present meta-analysis is reported following the Preferred Reporting Items for Systematic Reviews and Meta-Analyses of Diagnostic Test Accuracy studies (PRISMA-DTA declaration) [25] and is registered on PROSPERO under the number CRD42024502830.

### Study selection criteria

Our eligibility criteria were defined according to population, exposure, comparison, outcomes, and study type (PECOS). We included studies involving patients exposed to ARD who developed MRONJ after oral surgery. Oral surgery included tooth extraction, implants, and alveoloplasty.

Eligible studies were required to have data on sCTX levels available as the index test whether individuals or pooled. We excluded studies where sCTX levels of patients with spontaneous MRONJ were mixed with those experiencing MRONJ after oral surgery.

Patients exposed to ARD therapy who underwent an oral surgery without developing MRONJ were considered as comparators.

Any prospective and retrospective studies, regardless of the study design, were eligible. We also consider case series when patients ≥ 2. We did not consider in vitro or animal studies, single case studies, comments, editorials, systematic reviews. We did not use any date restrictions. We only considered studies published in English.

### Information sources, literature search, and study selection

A systematic search of the literature was conducted in MEDLINE (via PubMed), EMBase, and the Cochrane Library from inception to September 30, 2023. We applied specific search strategies tailored to each database. All search strategies are detailed in S1 File. Two authors (C.G., T.P.) independently screened the identified records for eligibility. Any disagreements between the authors were discussed reaching consensus, and if consensus could not be reached, a third author (P.L.) was invited for the final decision.

### Data collection

We extracted data regarding study design, demographics, treatment indication (osteoporosis versus cancer), drug characteristics (molecules, dosage, mode of administration), and sCTX levels (mean, standard deviation [SD], median, interquartile range [Q1, Q3], individual levels when available and assay method) using pre-designed forms.

### Risk of bias assessment

For each of the included studies, we independently and in duplicate assessed risk of bias on the basis of the results obtained, using the Newcastle-Ottawa Quality Assessment Form for Cohort Studies and Case Control Studies [26], and converted the scores to AHRQ standards (good, fair, and poor).

## Synthesis of results and meta-analysis

We performed a descriptive analysis of all eligible studies. In each study, patients were categorized based on their demographic characteristics, disease, and treatment. Weighted means for the median age, mean sCTX levels, and median sCTX levels were determined, according to the number of patients in each study. For quantitative synthesis, we assessed sCTX thresholds using percentile meta-analysis and descriptive statistics.

To assess the sCTX threshold beyond which the risk of developing MRONJ following oral surgery is minimal, we applied various analysis methods. We calculated the 95th, 97.5th, and 99th percentiles when individual data were available. In cases where only pooled data were available, the same percentiles were estimated from the mean and standard deviation on the assumption that the distribution of the data followed a normal distribution. As there is to our knowledge, no validated method of percentile meta-analysis in the current literature, we applied therefore, the most appropriate approach, i.e., a crude means meta-analysis (based on a single sample) where the mean was replaced by the percentile. The outcome of the meta-analysis therefore reflects the weighted average of the percentiles of each study. To facilitate this calculation, the standard deviation of the percentile was bootstrapped from individual sCTX data. For studies providing pooled data, all percentiles from the 1st to the 99th were estimated based on the mean and standard deviation, on the assumption of a normal distribution. These estimates were utilized to fictitiously portray the individual sample data. Subsequently, the standard deviation of the percentile was estimated through bootstrapping.

Meta-analyses were carried out to assess the overall result of the 95th, 97.5th and 99th percentiles, using the standard deviation of the percentile concerned. These three percentiles were calculated for exploratory purposes. However, our primary aim was to establish the safety threshold, thus defined by the 99th percentile, representing the point beyond which 1% of patients treated with ARD therapy experience MRONJ following oral surgery. A random-effect model with weighting by the method of DerSimonian and Laird was used for this purpose [27]. Meta-analyses of subgroups of studies according to the drug (bisphosphonates and denosumab) and to the bisphosphonates route of administration (oral and intravenous) were also planned, using the same method. Due to the small number of studies, the exploratory nature of the analysis and the various sources of error induced by the estimation and calculation methods used, no additional analysis of heterogeneity was planned.

Finally, when individual sCTX data were available, they were grouped and described in terms of mean and standard deviation, median and interquartile range, minimum and maximum values, and 95th, 97.5th and 99th percentiles. The number of subjects with sCTX levels above 150, 200, 300 and 400 pg/mL was also calculated. This descriptive analysis was also applied to subgroups according to treatment indication (osteoporosis and cancer). A Shapiro-Wilk test was performed to test the normality of the data distribution [28].

The various analyses were performed using RStudio software (v. 4.2.0).

## Results

### Search results and study selection

The initial database search yielded a total of 250 records, comprising 96 from PubMed, 150 from Embase, and 4 from the Cochrane Library. Following the removal of 73 duplicates, a comprehensive review of the titles and abstracts of the remaining 177 studies was conducted. During this selection phase, 131 records were excluded, resulting in 46 records that underwent a full article review for eligibility. One duplicate was eliminated after full reading of the articles (same article but different title and journal). The Fig 1 provides a full description of the selection process. A total of 7 studies were included in this analysis [25].

### Study and patients characteristics

The included studies consisted of 2 prospective cohort studies, 4 retrospective cohort studies, and 1 case-control study, totaling 7 observational studies, all meeting the predetermined eligibility criteria for both qualitative and quantitative synthesis, including a total of 1,281 patients (S1 Table) [22,29–34].

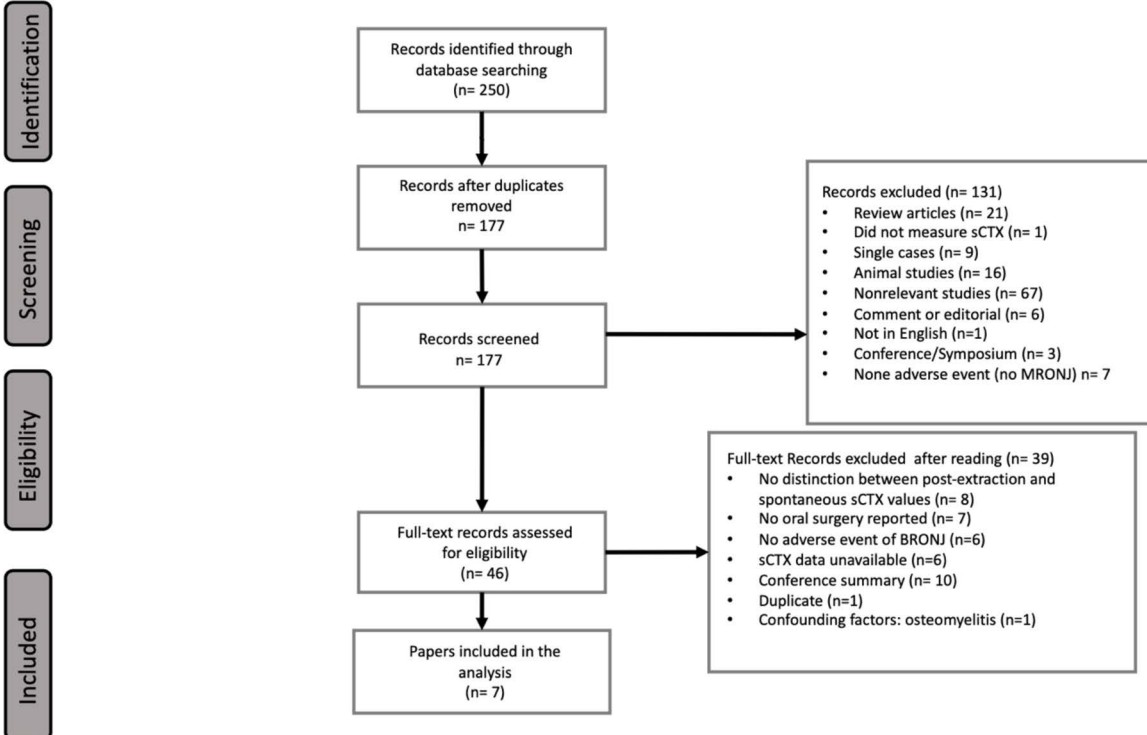

**Fig 1. PRISMA flow diagram.**

Among them, 5 studies provided individual data, including sCTX level, indication for ARD therapy, molecules and dosage administered, and attributed cause of osteonecrosis (n = 58) [29–33]. The other two studies presented pooled data concerning 1223 patients [22,34]. We contacted the original authors to obtain individual data but were unsuccessful.

The only ARDs studied were bisphosphonates, predominantly administered orally (94.5%). No study reported cases of patients treated with denosumab. Four studies had control groups (n = 1151), but with no data available regarding sCTX (individual data or pooled data) [22,30,33,34]. Thus, planned analyses regarding patients treated with denosumab, intravenous bisphosphonates, and comparisons versus controls could not be performed.

The assessment of studies regarding the risk of bias is summarized in S2 Table. Overall, the quality of the studies was poor for 3 studies [29,31,32], fair for 2 [30,35] and good for 2 [22,33].

Among the 1281 patients included in the 7 trials, 130 had experienced MRONJ, and 1151 were considered controls, i.e., patients exposed to antiresorptive drugs who had undergone dental surgery without developing MRONJ. However, we excluded 17 patients from the 130 with MRONJ for the following reasons: in 7 cases, osteonecrosis of the jaw had occurred spontaneously in the absence of oral surgery, 4 had sCTX missing data and 6 patients were identical in Kwon's studies of 2009 and 2011 [29,30].

Finally, we analyzed data from 1,264 patients, all of whom had undergone oral surgery. They were treated with bisphosphonates, primarily for osteoporosis (96.3%) and were mainly female (79.9%). The cancer-related conditions were primarily bone metastases and myeloma. On average the median age was 69.4 years (range: 20–99). On average the mean and median sCTX were 133.4 pg/mL and 122.4 pg/mL respectively with four missing data. Among them, 113 experienced MRONJ. The results of weighted average of mean and median sCTX according to the route of administration and bisphosphonate indication are listed in Table 1.

**Table 1. Weighted average of mean and median sCTX according to route of administration and bisphosphonate indication.**

| N total = 1190 * | N (%) | sCTX (Weighted average of mean), pg/mL | sCTX (Weighted average of median), pg/mL |
|---|---|---|---|
| Oral bisphosphonates | 1170 (98.3%) | 122.0 | 120.6 |
| Intravenous bisphosphonates | 20 (1.7%) | 261.7 | 224.0 |
| Osteoporosis indication** | 1170 (98.3%) | 122.0 | 120.6 |
| Cancer indication** | 20 (1.7%) | 261.7 | 224.0 |

*Data were unavailable for 74 patients.

**Among the 1190 patients with available data, oral bisphosphonates were always prescribed to osteoporotic individuals, whereas intravenous bisphosphonates were exclusively administered to those with cancer-related conditions.

Data regarding the specific types of molecules used and their dosages were only available in 3 studies (n = 180 patients). The most commonly used bisphosphonates were alendronate (65.0%) and risedronate (17.8%), with dosages consistent with approved indications [22,29,30].

## Meta-analysis

The first meta-analysis was conducted within the entire population of patients with MRONJ, including both cancer-related and osteoporotic patients (n = 113). The sCTX levels for the 95th, 97.5th, and 99th percentiles were 338.0 pg/mL [95%CI: 190,3; 485,7], 401.9 pg/mL [95%CI: 191,3; 612,6], and 458.0 pg/mL [95%CI: 190,4; 725,6], respectively (Fig 2), suggesting that among the entire population of patients treated with bisphosphonates and who developed MRONJ after oral surgery, only 1% had an sCTX level exceeding 458 pg/mL. The heterogeneity in the meta-analysis was significant ($I^2 = 100\%$).

The second meta-analysis was conducted within the population with individual data available (n = 58) and was restricted to the population of patients treated with oral bisphosphonates for osteoporosis who developed MRONJ (n = 38). The sCTX levels for the 95th, 97.5th and 99th percentiles were 185.3 pg/mL [95%CI: 131,3; 239,3], 187.4 pg/mL [95%CI: 133,9; 240,8] and 188.6 pg/mL [95%CI: 135,4; 241,9], respectively (Fig 3). The heterogeneity among the 4 studies was significant ($I^2 = 98\%$).

## Analysis of individual data

**sCTX Distribution in the population with available individual data.** Individual data were available for 58 patients with MRONJ after oral surgery. Their mean ± standard deviation and median (IQR) sCTX level were 157.7 ± 173.8 pg/mL and 116.0 pg/mL (70.0–192.0). The sCTX 95th, 97.5th and 99th percentiles were 393.0 pg/mL 481.0 pg/mL and 803.0 pg/mL respectively.

The distribution of sCTX values is illustrated in Fig 4a and did not follow a normal distribution (p-value <0.001). The number of patients with sCTX values exceeding 150, 200, 300, and 400 pg/mL was 19, 13, 4, and 3, respectively.

We performed a subgroup analysis based on the indication: osteoporosis (n = 38) and cancer-related indications (n = 20). sCTX values according to the bisphosphonate indication are summarized in Table 2, and their distribution is illustrated in Fig 4b and 4c.

In the group of patients treated for osteoporosis, the 99th percentile of sCTX was 260.0 pg/mL, with 7, 2, 0, and 0 patients having sCTX values exceeding 150, 200, 300, and 400 pg/mL, respectively. In contrast, among patients treated for cancer, the numbers were 12, 11, 4, and 3, respectively.

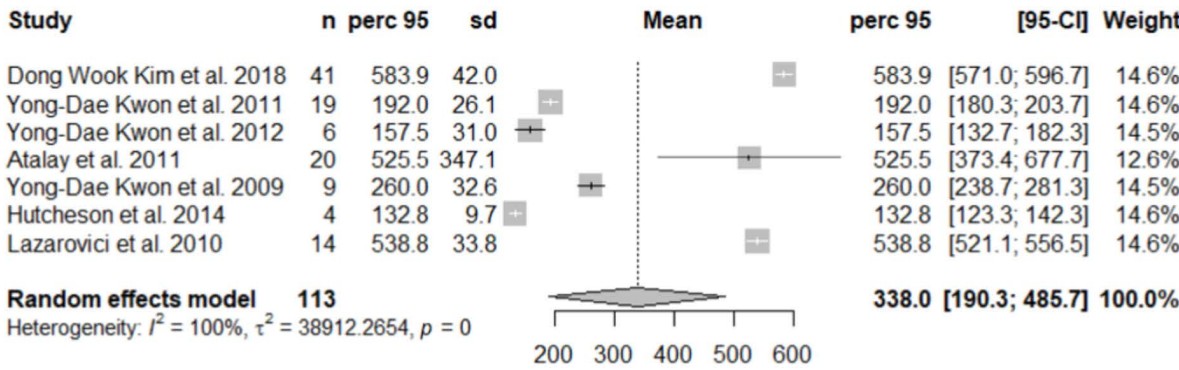

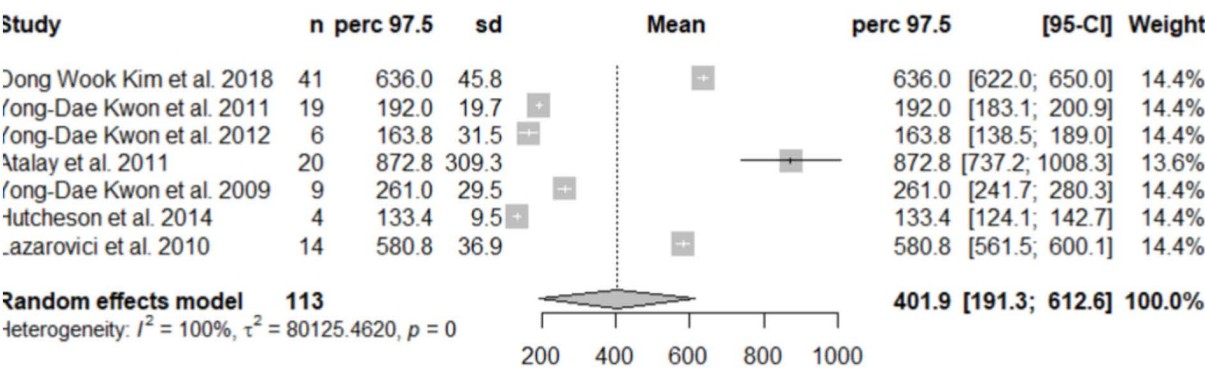

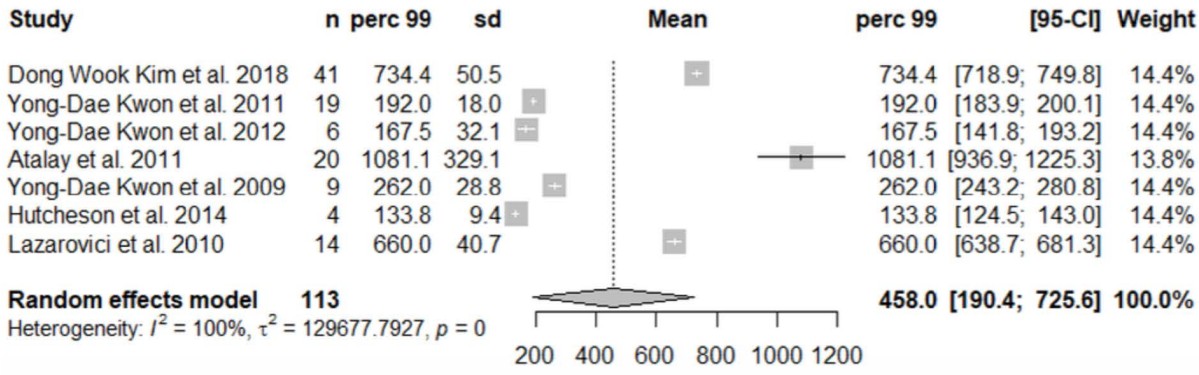

MRONJ: medication-related osteonecrosis of the jaw ; sCTX : serum C-terminal telopeptide of type I collagen; SD: standard deviation

**Fig 2. Meta-analysis of the 95th, 97.5th and 99th percentiles of all the included patients with MRONJ.**

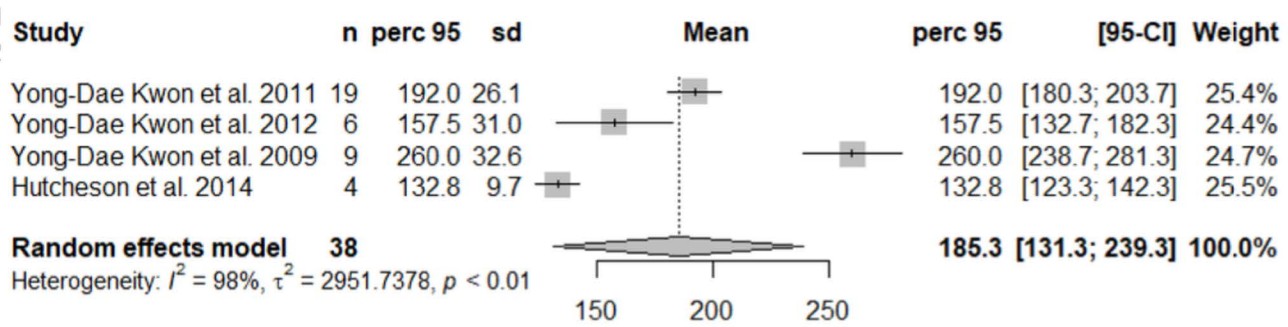

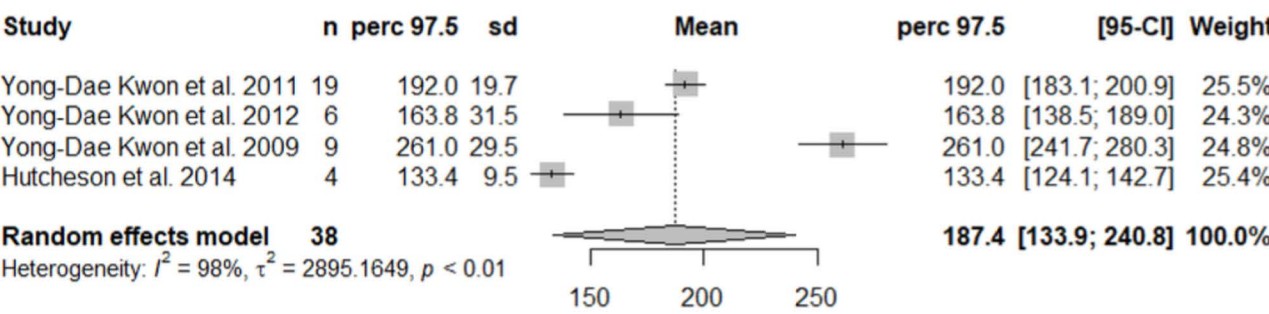

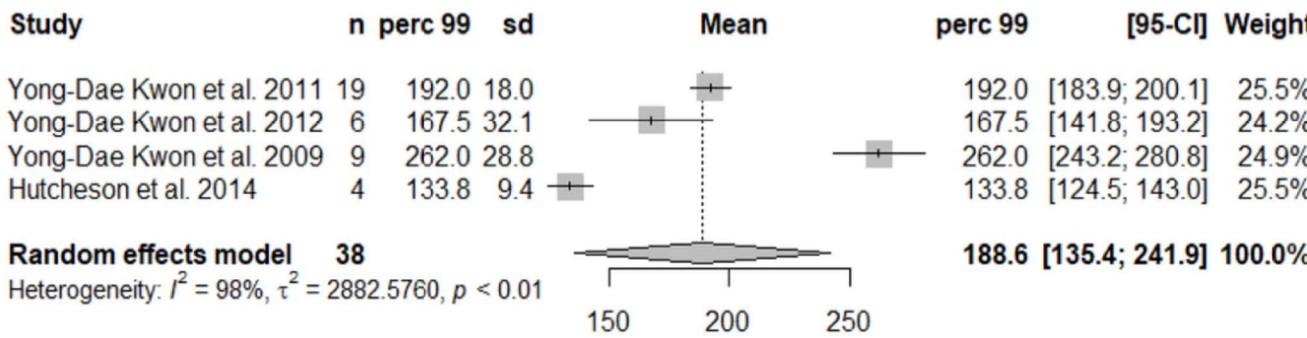

MRONJ: medication-related osteonecrosis of the jaw ; sCTX : serum C-terminal telopeptide of type I collagen; SD: standard deviation

**Fig 3. Meta-analysis of the 95th, 97.5th and 99th percentiles of the patients with MRONL treated for osteoporosis.**

a

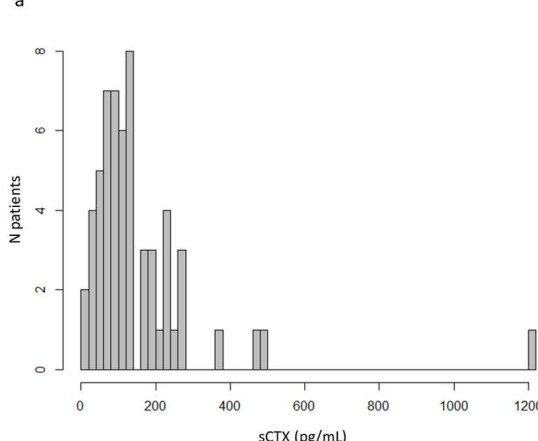

b

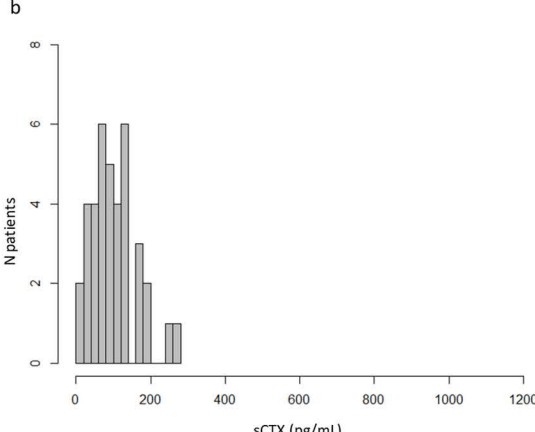

c

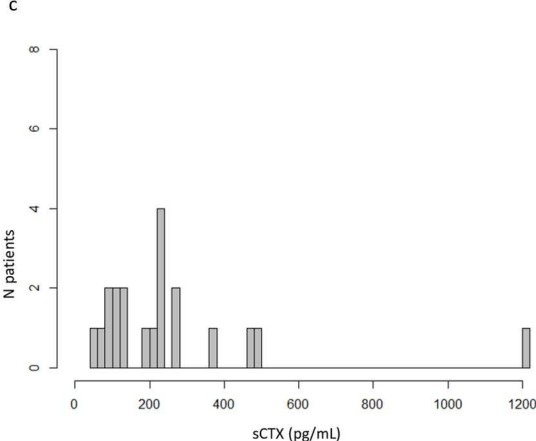

**Fig 4. Bar chart of sCTX levels of all individual values within a) the entire population with MRONJ (n = 58), b) patients treated for osteoporosis (n = 38), c) patients treated for cancer-related conditions (n = 20).** There is a significant dispersion of values within the population of patients with cancer-related conditions, which extends to the entire study cohort. This variability hinders the establishment of a safety threshold against medication-related osteonecrosis of the jaw (MRONJ) in these patients. In the osteoporotic patient group, only two individuals displayed sCTX levels above 200 pg/mL, with none exceeding 270 pg/mL.

**Table 2. sCTX values among patients treated with bisphosphonates who developed MRONJ after oral surgery, with individual data available.**

| sCTX values (pg/mL) | n | mean ± standard deviation | median (IQR) | 95th percentile | 97.5th percentile | 99th percentiles |
|---|---|---|---|---|---|---|
| Total | 58 | 157.7 ± 173.8 | 116.0 (70.0-192.0) | 393.0 | 481.0 | 803.0 |
| Osteoporosis indication | 38 | 102.9 ± 60.6 | 98.0 (63.0-132.0) | 202.0 | 257.0 | 260.0 |
| Cancer indication | 20 | 261.8 ± 257.1 | 224.0 (115.0-265.0) | 526.0 | 873.0 | 1081.0 |

## Discussion

Our meta-analysis aimed at determining an sCTX threshold above which the risk of osteonecrosis of the jaw after oral surgery in patients treated with bisphosphonates was negligible, provides three key findings. Firstly, the situation varies significantly depending on the indication for which bisphosphonates are prescribed. In osteoporosis, the estimated incidence of MRONJ following tooth extraction is 0.15%, while in cancer-related conditions, it ranges between 1.6% and 14.8% [14,35–38]. Our findings indicate that in cases of cancer, sCTX levels are not reliable biomarkers for predicting MRONJ. The 99th percentile of sCTX was found to be at 1081 pg/mL, with a distribution revealing levels of sCTX exceeding 1200 pg/mL, also observed in other samples [31]. These data confirm that the use of sCTX for predictive purposes of MRONJ in patients treated with bisphosphonate for cancer is not warranted [14]. Indeed, in this oncological context, sCTX levels not only reflect the action of bisphosphonates but, more importantly, also reflect the disease activity.

They have been shown to be effective markers of the progression of bone metastases or myelomatous involvement [39].

The second key finding is that conversely sCTX may be an appealing safety biomarker for predicting a low risk of MRONJ in patients treated with oral bisphosphonates for osteoporosis. We utilized two methods to determine the 99th percentile of sCTX in this population, meta-analysis and individual patient data analysis. In the meta-analysis, the 99th percentile was calculated within each study, followed by a weighted average of these results. For individual patient data, all sCTX values from all studies were pooled, sorted in ascending order, and the 99th percentile was calculated from this distribution of data, without considering methodological differences across studies. The thresholds determined by these two methods were 188 and 260 pg/mL, respectively. Therefore, adopting the most conservative approach, we can consider that when sCTX levels are beyond 260 pg/mL, the risk is very low. This threshold is supported by the descriptive analysis of individual patient data, where only 2 osteoporotic patients who experienced MRONJ had sCTX levels exceeding 200 pg/mL, and none had levels beyond 300 pg/mL.

The third key point is that, regardless of the indication for bisphosphonate use, the historically proposed sCTX threshold of 150 pg/mL is not adequate for assessing a low risk of MRONJ [24]. Our results are consistent with those of previous meta-analyses [40–44]. In these meta-analyses, the sensitivity and specificity of the 150 threshold did not exceed 57% and 77%, respectively [41,43].

In our population of patients who experienced MRONJ after oral surgery, on average the median and mean levels of sCTX were 122.4 and 133.4 pg/mL, respectively, confirming that a threshold of 150 pg/mL would hardly discriminate between patients.

The strength of this study lies in its use of an original method of percentile meta-analysis to determine the sCTX threshold associated with the lowest risk of MRONJ. But this approach also introduces certain limitations. To our knowledge, no validated method of percentile meta-analysis exists. For this purpose, we derived percentiles from the mean or standard deviation assuming a normal distribution and calculated their standard deviation using the bootstrap method based on individual sCTX data. However, the Shapiro-Wilk test revealed a non-normal distribution of the individual sCTX data. This accounts for the significant variation in sCTX values observed among the 95th (sCTX = 393 pg/mL), 97.5th (sCTX = 481 pg/

mL), and 99<sup>th</sup> percentiles (sCTX = 803 pg/mL). This variability was primarily attributed to an extreme and possibly aberrant value, which may also be a source of error.

Our percentile meta-analysis was conducted akin to a mean meta-analysis, with standard deviation calculation of the percentile derived from individual-level data, yielding simulated data which is not perfectly reliable. Nevertheless, this method enabled the computation of sCTX safety thresholds despite the limited data available in the literature and the results of the percentile meta-analyses were compared with individual data analyses. Whether it was the meta-analysis or descriptive analysis, the results were consistent.

The findings from the individual data analysis require careful interpretation. These data were pooled from studies employing varying methodologies, such as differing inclusion criteria, sCTX collection times, and assay techniques. CTX levels may not have been consistently measured at the same time, which could introduce variability due to the circadian rhythm's influence on sampling. Even if it is recommended that blood samples should be taken in the morning, none of the studies reported the timing of sampling or whether the patients were fasting. There was also a lack of standardization in the methodologies used across different laboratories [45–48].

Additionally, variations were noted in the timing of sCTX measurements, with some taken preoperatively and others at the time of BRONJ diagnosis.

Another limitation arises from the fact that we did not conduct heterogeneity analysis due to the low number of studies, small sample sizes, and potential sources of error in the calculations mentioned above.

Meta-analysis is a powerful tool for synthesizing evidence from multiple studies, providing increased statistical power, but it is constrained by the quality and heterogeneity of the included studies. Therefore, we conducted individual data analyses when possible, as this approach provides more granular, tailored insights with direct control over the data. Meta-analysis is a powerful tool for synthesizing evidence from multiple studies, providing increased statistical power, but it is constrained by the quality and heterogeneity of the included studies. Therefore, we conducted individual data analyses when possible, as this approach provides more granular, tailored insights with direct control over the data.

Taking these limitations into account, our results suggest that in patients treated with bisphosphonates for osteoporosis, an sCTX threshold above 260 pg/mL regardless of the assay method used or the time of sampling, is associated with minimal risk of MRONJ after oral surgery. This threshold could also guide potential treatment holiday prior to surgical intervention [49,50]. The various studies included in the meta-analysis reported little to no information on known established risk factors, such as smoking or oral hygiene, suggesting that the determined threshold would be relevant regardless of these risk factors [19,51]. These results need to be confirmed particularly with sCTX data from a population of control patients.

Our data also suggest that sCTX is not a reliable biomarker for identifying this risk in patients treated with bisphosphonates for cancer-related conditions. The risk is significant, regardless of the sCTX level. Although cancer is a well-known risk factor for MRONJ, prior to our work, the various reviews that sought to determine an sCTX threshold and considered it an unreliable turnover marker for predicting MRONJ did not differentiate between treatment indications or routes of administration in setting the threshold. While we confirm, in line with previous reviews, that sCTX is not a reliable marker for predicting MRONJ in cancer-related conditions, our results indicate that it may serve as a weak marker in osteoporosis.

## Supporting information

**S1 Table. Characteristics of the 7 included studies.**
(DOCX)

**S2 Table. Modified Newcastle-Ottawa Quality Assessment Scale (NOS) for cohort studies (Table 2a) and case control studies (Table 2b) applied to the studies included in the meta-analysis.**
(DOCX)

**S1 File. Electronic search strategy.**
(DOCX)

**S2 File. Table of all data extracted translated version.**
(XLSX)

## Author contributions

**Conceptualization:** Camille Ghio, Robinson Gravier-Dumonceau, Pierre Lafforgue, Roch Giorgi, Thao Pham.

**Data curation:** Camille Ghio.

**Formal analysis:** Camille Ghio, Robinson Gravier-Dumonceau, Thao Pham.

**Investigation:** Camille Ghio, Thao Pham.

**Methodology:** Camille Ghio, Robinson Gravier-Dumonceau, Roch Giorgi, Thao Pham.

**Project administration:** Camille Ghio, Pierre Lafforgue, Roch Giorgi.

**Resources:** Camille Ghio.

**Software:** Camille Ghio.

**Supervision:** Robinson Gravier-Dumonceau, Pierre Lafforgue, Thao Pham.

**Validation:** Camille Ghio, Robinson Gravier-Dumonceau, Pierre Lafforgue, Roch Giorgi, Thao Pham.

**Visualization:** Pierre Lafforgue, Roch Giorgi, Thao Pham.

**Writing – original draft:** Camille Ghio.

**Writing – review & editing:** Robinson Gravier-Dumonceau, Pierre Lafforgue, Roch Giorgi, Thao Pham.

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
