## [Decision Letter · Decision Letter 0]

5 Aug 2024

PONE-D-24-22980Identifying a Safety Threshold of Serum C-terminal telopeptide to Prevent Medication-Related Osteonecrosis of the JawPLOS ONE

Dear Dr. Pham,

Thank you for submitting your manuscript to PLOS ONE. After careful consideration, we feel that it has merit but does not fully meet PLOS ONE’s publication criteria as it currently stands. Therefore, we invite you to submit a revised version of the manuscript that addresses the points raised during the review process.

It is an interesting manuscript, please review comments and modify manuscript as appropriate. Thank you.  

We look forward to receiving your revised manuscript.

Kind regards,

Sian Yik Lim

Academic Editor

PLOS ONE

Journal Requirements:

2. Please identify your study as "systematic review and meta-analysis" in the title of your manuscript.

3. We note that there is identifying data in the Supporting Information file "Supplementary table 1 and 2". Due to the inclusion of these potentially identifying data, we have removed this file from your file inventory. Prior to sharing human research participant data, authors should consult with an ethics committee to ensure data are shared in accordance with participant consent and all applicable local laws.

-Location data

Additional Editor Comments:

This is an interesting study that tries to study the threshold of serum C-terminal telopeptide that predicts low risks of jaw osteonecrosis in patients receiving bisphosphonates. The authors reviewed the literature identified studies to investigate this interesting clinical topic, and pooled the cases. The results show that patients who are on oral bisphosphonates developed MR ONJ and had a 95-99 percentile range of serum CTX of 185-188. Individual data analysis showed 95 to 99% as 202-260. Based on their study, they recommend a threshold of 260 pg./mL for determining the low risk of developing MRONJ while on oral bisphosphonate treatment.

It is an interesting article but recommend that the authors address the following:

1. Did the authors assess for other risk factors that may affect serum CTX levels and the occurrence of MRONJ, such as comorbidities and length of bisphosphonate use?

2. Can the authors describe more about how they derived the patients included in the analysis of individual data, what cases were excluded and included, and whether it was 58 or 38 cases? It is confusing to read on page 11, paragraph 12, that Individual data were available for 58 patients with MRONJ after oral surgery, but then in the caption for figure 4, just below that paragraph, n=38.

3. In the discussion, an the authors discuss the possibility of other biomarkers that could be predictive of MR ONJ besides serum CTX, which is the reason why serum CTX is being particularly studied?

4. Can the authors include citations that serum CTX measurement may not be reliable and can be variable depending on the time of day the sample was obtained and how it was measured?

5. While 260 is a relatively reasonable number based on this study, due to various limitations, such as the large variability of the studies and the possibility of bias, I think the authors should acknowledge the limitations more, especially in regard to the variability of assays, measurement, etc. More studies are needed to determine if 260 is an appropriate threshold.

6. For the results in table 1, why do the authors think IV bisphosphonates have a higher level of weighted average mean of CTX than oral bisphosphonates. Theoretically it should be the other way around. Was IV mainly used in cancer patients?

7. Can the authors put their findings in the context of recent review that states that we do not have reliable boneturnover markers to predict risk of MRONJ

https://doi.org/10.1016/j.ajoms.2023.09.001Get rights and content

Reviewers' comments:

Reviewer's Responses to Questions

**Comments to the Author**

1. Is the manuscript technically sound, and do the data support the conclusions?

Reviewer #1: No

2. Has the statistical analysis been performed appropriately and rigorously? 

Reviewer #1: I Don't Know

3. Have the authors made all data underlying the findings in their manuscript fully available?

Reviewer #1: Yes

4. Is the manuscript presented in an intelligible fashion and written in standard English?

Reviewer #1: Yes

5. Review Comments to the Author

Reviewer #1: Thank you for your diligent efforts in identifying a predictive marker for MRONJ. The article is highly engaging and informative. However, I have some concerns as outlined below:

Title

Since the "safety threshold" was not directly established in this study, referring to the "predictive sCTX level" would be more appropriate.

The short title is quite general and lacks specific information. It would be beneficial to make it more informative to better convey the study's focus.

Abstract

Result:

- “Among those treated with oral bisphosphonates for osteoporosis (n = 58),……”

The data from Table 2 shows the number of MRONJ patients having bisphosphonate for osteoporosis is 38, not 58, is this number correct?

Conclusion:

- This study did not provide sCTX levels for patients who did not develop MRONJ, making it impossible to establish the proposed sCTX level as a definitive threshold. Without this comparison, it remains unclear whether patients who have sCTX levels higher than the proposed threshold will not develop MRONJ.

Main text

Introduction:

To enhance the article's impact, it is crucial to add more issues of MRONJ.

Result:

Study and Patients Characteristics:

- “The 1264 patients analyzed had all undergone oral surgery. They were treated with bisphosphonates, primarily for osteoporosis (96.3%)”

Is this data from the same analysis as those in Table 1?

- “Table 1. Weighted average of mean and median sCTX according to route of administration and bisphosphonate indication of the patients with a MRONJ following oral surgery.”

I noticed that your article states the number of patients who developed MRONJ following oral surgery was 113, not 1,190. Please verify the accuracy of the data in this table and ensure clarity.

- To make the data more informative, the mean and median sCTX levels for MRONJ and non-MRONJ patients should be reported separately rather than as an overall combined level.

Discussion:

- To make your discussion more compelling, discuss the strengths and weaknesses of meta-analysis and individual data analysis. This will demonstrate why the sCTX level from individual data analysis was proposed as the threshold level.

- The summary stating “our results suggest that in patients treated with bisphosphonates for osteoporosis, a sCTX threshold above 260 pg/mL is associated with minimal risk of MRONJ following oral surgery” cannot be conclusively drawn. This is because the sCTX levels of patients who did not develop MRONJ after oral surgery were not investigated. Therefore, it remains uncertain whether the sCTX levels of non-MRONJ patients are higher or lower than the proposed threshold.

6. PLOS authors have the option to publish the peer review history of their article (what does this mean? ). If published, this will include your full peer review and any attached files.

**Do you want your identity to be public for this peer review?** For information about this choice, including consent withdrawal, please see our Privacy Policy .

Reviewer #1: **Yes: ** Keskanya Subbalekha.

---

## [Author Response · Author response to Decision Letter 0]

24 Oct 2024

Dear Editor,

The responses to the reviewers' questions and comments have been uploaded in the document titled "Response to Reviewers"

Kind regards,

---

## [Decision Letter · Decision Letter 1]

19 Nov 2024

PONE-D-24-22980R1Identifying a Threshold of Serum C-terminal telopeptide with a low risk of Medication-Related Osteonecrosis of the Jaw secondary to oral surgery: A systematic review and meta-analysisPLOS ONE

Dear Dr. Pham,

Thank you for submitting your manuscript to PLOS ONE. After careful consideration, we feel that it has merit but does not fully meet PLOS ONE’s publication criteria as it currently stands. Therefore, we invite you to submit a revised version of the manuscript that addresses the points raised during the review process. Thank you for addressing the reviewers comments, please consider making changes per reviewer ones comments. 

We look forward to receiving your revised manuscript.

Kind regards,

Sian Yik Lim

Academic Editor

PLOS ONE

Journal Requirements:

Reviewers' comments:

Reviewer's Responses to Questions

**Comments to the Author**

1. If the authors have adequately addressed your comments raised in a previous round of review and you feel that this manuscript is now acceptable for publication, you may indicate that here to bypass the “Comments to the Author” section, enter your conflict of interest statement in the “Confidential to Editor” section, and submit your "Accept" recommendation.

Reviewer #1: (No Response)

2. Is the manuscript technically sound, and do the data support the conclusions?

Reviewer #1: Yes

3. Has the statistical analysis been performed appropriately and rigorously? 

Reviewer #1: I Don't Know

4. Have the authors made all data underlying the findings in their manuscript fully available?

Reviewer #1: Yes

5. Is the manuscript presented in an intelligible fashion and written in standard English?

Reviewer #1: Yes

6. Review Comments to the Author

Reviewer #1: Dear Authors,Thank you very much for your efforts; your responses have clarified my queries. I do still have a concern regarding the title, though, and would appreciate your input.

Title: Please consider the right word according to your study

The meanings of "threshold" and "predictive level" are different. "Threshold" implies a cut-off point, often associated with safety or effectiveness. For instance, in a clinical setting, a "threshold" typically means a specific value above or below which an intervention is indicated or a condition is considered more manageable. "Predictive level" refers to a marker that indicates the likelihood or probability of a certain outcome, without necessarily implying safety.

7. PLOS authors have the option to publish the peer review history of their article (what does this mean? ). If published, this will include your full peer review and any attached files.

**Do you want your identity to be public for this peer review?** For information about this choice, including consent withdrawal, please see our Privacy Policy .

Reviewer #1: **Yes: ** Assoc.Prof.Dr. Keskanya Subbalekha

---

## [Author Response · Author response to Decision Letter 1]

3 Jan 2025

We thank the reviewer #1 for this comment

We have change the title with “predictive levels” instead of “threshold”

---

## [Editor Report · Decision Letter 2]

14 Jan 2025

Identifying a Predictive Level of Serum C-terminal telopeptide Associated with a low risk of Medication-Related Osteonecrosis of the Jaw secondary to oral surgery: A systematic review and meta-analysis

PONE-D-24-22980R2

Dear Dr. Pham,

We’re pleased to inform you that your manuscript has been judged scientifically suitable for publication and will be formally accepted for publication once it meets all outstanding technical requirements.

Kind regards,

Sian Yik Lim

Academic Editor

PLOS ONE

---

## [Editor Report · Acceptance letter]

PONE-D-24-22980R2

PLOS ONE

Dear Dr. Pham,

I'm pleased to inform you that your manuscript has been deemed suitable for publication in PLOS ONE. Congratulations! Your manuscript is now being handed over to our production team.

Kind regards,

on behalf of

Dr. Sian Yik Lim

Academic Editor

PLOS ONE